# Molecular Aspects of Allergen-Specific Immunotherapy in Patients with Seasonal Allergic Rhinitis

**DOI:** 10.3390/cells12030383

**Published:** 2023-01-20

**Authors:** Marina Izmailovich, Yuliya Semenova, Gulzada Abdushukurova, Ainur Mukhamejanova, Azhar Dyussupova, Raida Faizova, Meruert Gazaliyeva, Leila Akhvlediani, Natalya Glushkova, Sundetgali Kalmakhanov, Geir Bjørklund

**Affiliations:** 1Department of Internal Diseases, Karaganda Medical University, Karaganda 100008, Kazakhstan; 2School of Medicine, Nazarbayev University, Nur-Sultan 010000, Kazakhstan; 3Department of Therapy, Faculty of Postgraduate Medical Education, Shymkent Medical Institute, Shymkent 160006, Kazakhstan; 4Department of Family Medicine No 2, Astana Medical University, Nur-Sultan 010000, Kazakhstan; 5Department of General Medical Practice of Semey City, Semey Medical University, Semey 071400, Kazakhstan; 6Vice-Rector for Clinical Work, Astana Medical University, Nur-Sultan 010000, Kazakhstan; 7School of Medicine & Health Sciences, BAU International University Batumi, 6010 Batumi, Georgia; 8Department of Epidemiology, Biostatistics & Evidence Based Medicine, Al-Farabi Kazakh National University, Almaty 050040, Kazakhstan; 9Department Health Policy and Organization, Al-Farabi Kazakh National University, Almaty 050040, Kazakhstan; 10Council for Nutritional and Environmental Medicine (CONEM), 8610 Mo i Rana, Norway

**Keywords:** seasonal allergic rhinitis, allergen-specific immunotherapy, molecular allergy diagnostics, component-resolved diagnostics, sensitization, cross-reactivity

## Abstract

A systematic review and narrative synthesis of publications was undertaken to analyze the role of component-resolved diagnosis technology in identifying polysensitization for the provision of allergen-specific immunotherapy to patients with seasonal allergic rhinitis. A search of publications was carried out in electronic databases in accordance with the Preferred Reporting Items for Systematic Reviews and Meta-Analyses guidelines. The search helped to identify 568 publications, 12 of which were included in this review. Overall, 3302 patients were enrolled. The major finding was that component-resolved diagnostics change the choice of relevant allergens for allergen-specific immunotherapy in at least 50% of cases. Sensitization to allergen components differs with age, type of disease, and overall disease duration. Patients who had both bronchial asthma and allergic rhinitis were sensitized to a larger number of allergens than patients who had bronchial asthma alone.

## 1. Introduction

Currently, seasonal allergic rhinitis (AR) presents a global health problem [1]. According to statistics from the World Health Organization (WHO), allergic disorders affect approximately 40% of the global population [2,3] and their incidence continues to grow with increasing disease severity and expanding the spectrum of sensitization to unrelated allergens [4].

Seasonal allergic rhinitis is one of the biggest challenges among all seasonal allergic disorders [3], and it currently affects over 500 million people worldwide [5,6]. According to estimates, AR is observed in 23–30% of Europe’s population, in 12–30% of the population of the USA, and 17–35% of the adult population in Russia [7,8,9]. The patients often present allergic reactions to plant food allergens, which might not be cross reactive, making the diagnostics even more complicated [10].

Pollinosis is perhaps the most common seasonal allergic disorder and is prevalent in approximately 30% of the world’s population [11]. Most frequently, pollinosis manifests as AR, accompanied by rhinorrhea, nasal congestion, itching in the nasal cavity, and sneezing, and these symptoms reverse after the termination of exposure or initiation of treatment [3,12]. The course of the disease might be deteriorated by many factors, to which belong specific climate conditions, including a poor ecological situation and the level of socioeconomic development [11]. If AR is not treated appropriately, it can transform into bronchial asthma, chronic sinusitis, eustachitis, nasal polyposis, serous otitis media, allergic conjunctivitis, and other disorders [6,13].

The symptoms of AR impact quality of life, deteriorate daily living and work performance, and are a reason for seeking medical care [14]. Approximately 50–80% of patients with AR report sleep disturbances provoked by the disease, resulting in daytime fatigue, decreased awareness, inability to concentrate, depression, anxiety, and irritability [15,16].

To date, allergen-specific immunotherapy (ASIT) is the only curative method available for the treatment of seasonal AR [17,18,19]. This method impacts all pathogenic mechanisms of the allergy, potentiates preventive effects, and enables lasting remission after the end of treatment [20,21]. According to the Allergic Rhinitis and its Impact on Asthma (ARIA) guidelines that were initiated at the WHO workshop, ASIT is a disease-modifying therapy [22,23] with grade IA; recommendation [24,25]. The clinical effects of ASIT include reduction of AR symptoms, decreased demand for antiallergic agents, prevention of the expansion of the allergen spectrum, and bronchial asthma in patients with seasonal AR. Finally, ASIT improves patient quality of life [26,27,28].

The method is based on the systematic introduction of causative allergens in gradually increasing dosages [29]. This leads to specific hyposensitization, which promotes immune tolerance to the causative allergen [30]. In keeping with international guidelines, ASIT has to be provided for three to five years to achieve stable results, although the meaningful effect is already visible after the first year of treatment [31,32]. At present, ASIT is available in two forms: subcutaneous immunotherapy (SCIT) and sublingual immunotherapy (SLIT) [33,34].

Allergen-specific subcutaneous immunotherapy was developed in 1908 and has since gained much recognition worldwide [35]. However, the application of allergen-specific sublingual immunotherapy caused many doubts due to unpurified vaccines, and it became widespread only within the past two decades [36,37]. SLIT has several advantages compared with SCIT [38]. First, it could be used in children from the age of five. Second, only the first intake of medication has to be made in the presence of a doctor, and other intakes could be made independently at home with only periodic visits to a doctor for a follow up. Third, when medication comes in tablet form, it causes less discomfort for a patient, in contrast to repeated injections, which may be quite stressful. In addition, SLIT has a better safety profile with only local side effects and an extremely low incidence of anaphylaxis and other life-threatening conditions, often associated with treatment discontinuation [39,40,41]. Nonetheless, when we focus on seasonal diseases, comparative data show that SCIT has a higher efficacy [39]. Thus, ASIT is currently considered the preferred treatment for allergic disorders [42] and has sufficient proof of clinical effectiveness [43].

Annually, the number of polysensitized patients grows [44,45], which complicates the provision of allergen-specific immunotherapy (ASIT) to all those who need it [18]. According to the latest estimates, polysensitization is present in over 80% of patients with AR [18,24], and it is a well-known fact that this category of patients has more severe clinical manifestations and impaired quality of life [18,46]. There are rather contradictory opinions regarding the effectiveness of ASIT in polysensitized patients [44]. The European Medicines Agency (EMA) recommends monovalent ASIT in patients with polysensitization, considering it a safe and effective treatment [47,48,49]. However, polyvalent immunotherapy is preferred for polysensitized patients in the USA due to its effectiveness and safety [47].

Component-resolved diagnostics (CRD) can make the diagnosis of major and cross-reactive sensitization more accurate, forecast the risks of both local and systemic reactions [26,27], and provide a personalized approach to the choice of optimal ASIT, which is particularly important for polysensitized patients [50,51]. The World Allergy Organization, as well as the European Academy of Allergy and Clinical Immunology (EAACI), published a consensus document devoted to molecular diagnostics with an exceptional focus on CRD as a tool for practicing allergists [52,53]. CRD has proven to be a reliable diagnostic instrument that is used for distinguishing between allergens that cause clinically significant allergic rhinitis and allergens associated with symptomless sensitization [54].

Sensitization to the allergen does not always witness allergy activity and severity, and thus, elucidating the causal allergen CRD is used, which helps to identify the major and minor allergens. When significant sensitization to major allergens is identified, one can conclude about the importance of this allergen and allergy severity [55].

The composition of causal allergens may change over time [56]. Since the immune response diminishes with aging, many patients note a reduction in the severity of allergy symptoms. However, sensitization to new allergens may occur, and the sensitization profile may expand due to the presence of comorbidities or a long duration of the allergy [57]. Nevertheless, it is worth noting that ASIT may prevent the appearance of new sensitization [58].

CRD determines objective criteria for the initiation of ASIT and helps to predict whether it will be effective [46]. Because ASIT is a long-lasting and resource-consuming treatment approach, precise diagnostics, appropriate selection of patients, and identification of major sensitizing allergens are needed to optimize treatment planning, including financial considerations [59,60]. Most patients receiving ASIT have elevated serum levels of allergen-specific IgE within the first few months after treatment initiation [61], which has no consequences in terms of recurrence or aggravation and presents a natural response to ASIT [62]. Progressive reduction of allergen-specific IgE is commonly observed after 6–12 months and corresponds with a decrease in the numbers of residential IgE-secreting long-lived plasma cells in the bone marrow [63]. Two weeks following the initiation of ASIT are accompanied by elevated serum levels of allergen-specific IgG4 and by the secretion of Il-10-producing regulatory B-cells [19,37]. The levels of IgG4 in blood serum remain elevated for more than a year [61,64]. The allergen-specific IgA produced during SLIT may be the source of highly effective blocking antibodies on the surface of mucous membranes [65]. Such blocking antibodies inhibit allergen-IgE binding, which leads to decreased basophil histamine release [61].

A relationship between specific IgE and the effectiveness of ASIT may be attributed to specific IgG, which is the marker of ASIT effectiveness. Shamji and coworkers reported a close correlation between IgG levels and the clinical response to ASIT. The levels of specific IgE and IgG were interrelated [66]. In general, specific IgE is used to monitor the effectiveness of immunotherapy, as its reduction reflects the development of immune tolerance [67]. IgE serum levels decrease and are maintained at low levels for years following successful ASIT [39,40]. Thus, it appears reasonable to suppose that the higher the IgE levels are, the more clinically significant the allergen for an individual patient [65], and the more causative allergens are used for ASIT, the more effective the results will be [68]. For this reason, baseline-specific IgE levels to allergens could serve as reliable biomarkers of effectiveness in patients with allergic rhinitis [23,69].

Thus, it is essential to understand what predictors impact the effectiveness of ASIT [70], and a careful assessment of allergen profiles with the identification of major and cross-reactive allergens is needed [71]. Component-resolved diagnosis (CRD) could help to solve this issue [25,26]. This review analyzes the role of molecular allergy diagnostics in identifying polysensitization for the provision of ASIT to patients with seasonal allergic rhinitis.

## 2. Materials and Methods

A comprehensive search of publications in electronic databases was carried out to compile this systematic review. The search was specifically focused on papers devoted to the prospects of CRD in identifying polysensitization in patients with seasonal allergic rhinitis. To meet this goal, the following databases and search engines were utilized: PubMed, Cochrane Library, Scopus, Publons, and Google Scholar. For our systematic review of the literature, we adhered to the guidelines set forth by the Preferred Reporting Items for Systematic Reviews and Meta-Analyses (PRIS-MA). The flow diagram of the selection of studies for inclusion in this review is presented in Figure 1.

### 2.1. Data Sources and Searches

In the first step of search queries, every publication was evaluated by its title, and in the second step, the publication’s abstract was assessed to decide if the study met the inclusion criteria. When the publication was considered suitable, the full text was carefully evaluated. While searching in the PubMed database (accessed on 1 June 2022), the following search criteria were applied: [“Immunotherapy” (MeSH)] or [“Desensitization, Immunologic” (MeSH)] and [“Rhinitis, Allergic, Seasonal” (MeSH)] and [“Polysensitization” (title/abstract;TIAB)] and [“Molecular allergy diagnostics” (MeSH)] and [“Component-resolved diagnostics” (title/abstract; TIAB)]. All searches in PubMed, Publons, Cochrane Library, and Scopus were limited to papers in English published between 1 January 2008 and 31 January 2021. In addition, we looked for papers and articles in Russian and thus addressed eLIBRARY (accessed on 1 June 2022) for this purpose.

### 2.2. Study Selection

Inclusion and exclusion criteria for study selection were set by three reviewers (M.I., Y.S., and N.G.) and are presented in Figure 1. After the selection of publications that met the eligibility criteria, the following data were extracted—(i) last name of the first author and year of publication; (ii) patient age; (iii) sample size; (iv) type of allergy; (v) country; (vi) score on Ottawa-Newcastle scale [27]; and (vii) the main findings.

Type of Participants: studies of patients without prior confirmation of the type of sensitization were not eligible for inclusion.

Type of Exposure: CRD confirms the choice of allergen indication and uses for ASIT.

Type of Outcome: the primary outcome was the decision on the correctness of the ASIT protocol and diagnostic accuracy for immunotherapy prescriptions. Secondary outcomes included identifying allergen source types, changing allergen source, and diagnostic concordance before and after CRD.

Study Design: observational studies, such as case reports and case series, case-control studies, randomized control trials, and prospective and retrospective cohort studies were eligible for inclusion.

### 2.3. Data Synthesis and Analysis

In total, 12 papers were considered suitable for inclusion (Table 1). Two coauthors (M.G. and G.A.) extracted the data presented in selected publications. Any differences in opinion were resolved in discussions between A.D., M.I., and R.F. Then, G.B., S.K., L.A., and A.M. drafted the initial version of this paper after considering all suitable papers. The final version was compiled based on feedback received from all coauthors and scores obtained on the Ottawa-Newcastle scale. Finally, the major findings on the place of molecular allergy diagnostics in the identification of polysensitization for the subsequent provision of ASIT were summarized, and the resulting conclusions were elaborated.

## 3. Results

The initial search conducted in PubMed resulted in 521 publications, while the search in other databases helped to identify 47 additional records. When all duplicates were removed, 568 papers remained, and these were subjected to a thorough check. After the selection of publications, 67 papers were considered eligible based on the inclusion criteria. Application of the exclusion criteria resulted in 12 publications, which were included in this systematic review. Table 1 presents a summary of data retrieved from these publications.

### 3.1. Patient Characteristics

Overall, the patient population included 3302 people. Patients were selected through a skin prick test and clinical history in seven studies [73,74,75,76,78,81,83], specific IgE and clinical history in one study [72], provocation tests and clinical history in two studies [77,82], and purely IgE sensitization or skin prick test in two studies [79,80]. Patients were monosensitized in three studies [77,78,79], polysensitized in two studies [81,82], and mono- and polysensitized in seven studies [72,73,74,75,76,80,83]. One study included only adult participants. Three studies were carried out on a pediatric population, while the remaining eight studies used both children and adults as subjects. The majority of publications were focused on the ability and applications of molecular diagnostics in the accurate selection of allergens for the subsequent provision of ASIT. The studies included a determination of the specific IgE against recombinant allergens of grass, trees, and house dust mites. Changes in the provision of ASIT protocols were analyzed following the application of CRD. According to the data, allergy diagnostics often result in a change in allergens relevant to ASIT.

### 3.2. Evaluation of Component-Resolved Diagnosis

CRD is a tool for the identification of IgE sensitization to a panel of allergens. CRD can distinguish between primary sensitization and cross-reactive sensitization by understanding the major allergens and cross-reacting molecules [72]. This makes the identification of primary sensitization more precise and is very important for the selection of ASIT and the evaluation of its effectiveness [73,74,78]. Inhalant allergens play a particular role in seasonal allergies, and pollen grains of different plant families are the most common inhalant allergens (Table 2).

### 3.3. Efficiency of Component-Resolved Diagnosis for Use in ASIT

The use of CRD impacts the choice of allergens for ASIT in at least 50% of cases [72,73,74,77,78,80,83]. This is of great importance for patients with polysensitization, as an accurate selection of allergen panels provides not only clinical benefits but also economic advantages [73,75,80,81,83]. CRD enables the detection of cross-reactive allergen components [72,79,82]. As a general rule, these are minor allergen components that are not used in ASIT. It was demonstrated that decreased levels of IgE could serve for monitoring allergen-specific blocking of non-IgE antibodies induced by ASIT [78].

### 3.4. Outcomes

The outcomes of studies included in this review were assessed on the basis of the diagnostic accuracy of CRD for immunotherapy prescriptions. Influence variables included allergenic molecule, marker type, and type of sensitization (monosensitization vs. polysensitization). According to the available evidence, these factors significantly influenced the change in the allergen composition of ASIT. CRD use in clinical practice leads to more detailed information on IgE reactivity at the molecular level that could be helpful in the diagnosis of a respiratory allergy and might help clinicians choose appropriate pollen extracts for ASIT (Table 1).

## 4. Discussion

This systematic review demonstrated the importance of using CRD in establishing a precise diagnosis [79,82] and identifying true and cross-reactive allergens [75,77], which ensures the correct selection of allergen composition for allergen-specific immunotherapy in children and adults [76,80,81]. By studying cross-reactive allergens, valuable information on potential sensitization to different sources of allergens and associated clinical symptoms is obtained. Some cross-reactive allergens may cause severe clinical presentation, while exposure to others will not lead to any symptoms [57,87]. The list of protein families causing cross-reactivity with inhalant allergens in seasonal allergic rhinitis is presented in Table 3.

The identification of specific IgE markers of true sensitization and cross-reactive molecules is important for CRD in patients with polysensitization [75]. Cross-reactive carbohydrate determinants (CCD) could be the source of positive IgE detection results with no clinical significance. In a serum-based allergy diagnosis, antibodies of the IgE class are directed against CCDs, therefore, give the impression of polysensitization. Anti-CCD IgE, however, does not seem to elicit clinical symptoms. Diagnostic results caused by CCDs are therefore regarded as false positives [55]. Some commercial tests, including CCD inhibitors such as ALEX, have been applied in clinical practice and have shown advantages in reducing false-positive IgE results without impacting diagnostic sensitivity toward relevant allergens [88].

The problem with polysensitized patients is that they are sensitized to multiple sources of allergens, including cross-reactive ones. In Moreno et al.’s study, 78.2% of the subjects had polysensitization, while the rest were monosensitized or had a negative result [75]. Hu et al. showed that more than 93.0% of subjects were sensitive to more than one allergen component [82]. Haidar et al. reported that the majority of patients were polysensitized (62.65%) [83]. Therefore, the question of prescribing multiple allergens for these patients with ASIT, and how effective this treatment will be, has long been discussed. Calderon et al. concluded that single-dose therapy with two unrelated allergens is clinically effective [44]. Darsow et al. reported that CRD would be helpful in deciding on the indication for conventional ASIT in polysensitized patients [77]. In addition, Movérare et al. revealed that CRD use in clinical practice leads to more detailed information on IgE reactivity at the molecular level that could be helpful in the diagnosis of a polyclonal allergy, and might help clinicians choose appropriate pollen extracts for ASIT [72].

The implementation of a component-resolved diagnosis utilizing a range of pollen allergens is necessary for more precise patient characterization. This will enable the selection of a suitable allergen immunotherapy product, thereby allowing for the personalized management of seasonal allergic rhinitis [72,73,74,76,77,78,80,81,82,83]. Many authors report the importance of using CRD to select and influence the effectiveness of ASIT. Moreno et al. argue that when selecting the composition of AIT based solely on the results of SPT (skin prick test), approximately one-third of the patients would be treated with an allergen to which they were not allergic. Thus, according to the results of CRD, more than half of the patients had a change in the composition of ASIT compared to the initial selection [75]. Sastre et al. observed that the composition of AIT before and after molecular diagnosis coincided only in 46% of patients [73]. Letrán et al. reported a change in the ASIT protocol in more than 50% of cases after using CRD [74]. Moreover, Del-Río Camacho et al. described in their study that protocols for ASIT were changed in 54.3% of cases after CRD was conducted [81]. Martinez-Cañavate et al. reported that specialists changed the composition of the prescribed immunotherapy in 52.87% of cases [80]. In addition, Schmid et al. found that CRD might be a useful companion diagnostic to monitor the efficacy of SCIT [78]. These facts confirm the need to use CRD when selecting the composition of allergens for AIT.

High effectiveness of ASIT could only be expected in patients having IgE-antibodies to major allergens [89]. If some pollen allergen has several major allergen components, the extract for ASIT should include all of these components to enable sufficient effectiveness. ASIT will likely be moderately effective if a patient presents with a specific IgE to both major and minor allergens. Meanwhile, the absence of IgE to major allergens suggests a low response to ASIT [84].

Identification of potential ASIT patients with typical symptoms should begin at primary health care centers and include a detailed allergy history, medical examination, and assessment of serum total IgE levels to confirm allergic sensitization. In the next step, specific allergy tests should be carried out to elucidate the causes of seasonal AR. For this purpose, skin prick tests and specific IgE antibodies could be performed. When a diagnosis of seasonal allergic rhinitis is confirmed, a patient is better referred for specialized allergy care to make further tests and ASIT, which is carried out at the remission stage. In addition to a preliminary evaluation of symptoms, disease severity, and quality of life, the set of allergens for ASIT need to be selected by CRD technology. According to the European Academy of Allergy and Clinical Immunology (EAACI) guidelines, total and specific IgE should be used as biomarkers of ASIT effectiveness [50]. The summarized version of EAACI recommendations is presented in Figure 2 [50,90,91].

Medical professionals can clearly understand the diagnostic trajectory of individuals with suspected allergic rhinitis by using this algorithm. Furthermore, it allows for the reduction of overdiagnosis errors during the primary medical care stage. This algorithm also serves as a guide for allergology experts, as it incorporates current advice on cutting-edge technology for allergy diagnosis and treatment. Besides, the algorithm enables differential diagnosis of seasonal allergic rhinitis on the basis of clinical and laboratory criteria for an optimal choice of allergen-specific immunotherapy. The algorithm reflects how the obtained data can help to monitor the effectiveness of ASIT with respect to laboratory findings and also considers the assessment of symptom severity and quality of life.

The review’s strengths are that we investigated the impact of several variables on treatment outcomes, including allergenic molecule type, marker type, and sensitization type, and discovered that they did have a significant impact on the ASIT protocol. This study describes the allergen’s major and cross-reactive components at the molecular level, which can aid in the diagnosis of a respiratory allergy and help clinicians select appropriate pollen extracts for ASIT.

This review has several limitations. The most significant limitations are financial, as CRD is an expensive method of research that may not be available to all patients. A limitation of some studies is the small number of participants, different sensitization profiles, and a different disease profile when allergic rhinitis is combined with bronchial asthma. In addition, different age profiles were included in some of the studies.

## 5. Conclusions

Modern allergology services prioritize the reduction of seasonal allergic rhinitis rates. CRD is highly relevant in this regard for the diagnostic accuracy of CRD for immunotherapy prescription. Such an integrated approach is most appropriate for assessing major and cross-reactive allergen components as well as identifying cases of polysensitization. Furthermore, the search for prognostic biomarkers that can predict the efficacy of ASIT must be continued. Thus, identifying reliable markers enabling the evaluation of treatment response and the prediction of outcomes in patients with different sensitization profiles would be highly desirable. Finally, it appears necessary to keep up with the development of new approaches to allergy diagnostics for choosing the most important parameters influencing the treatment strategy.

## Figures and Tables

**Figure 1 cells-12-00383-f001:**
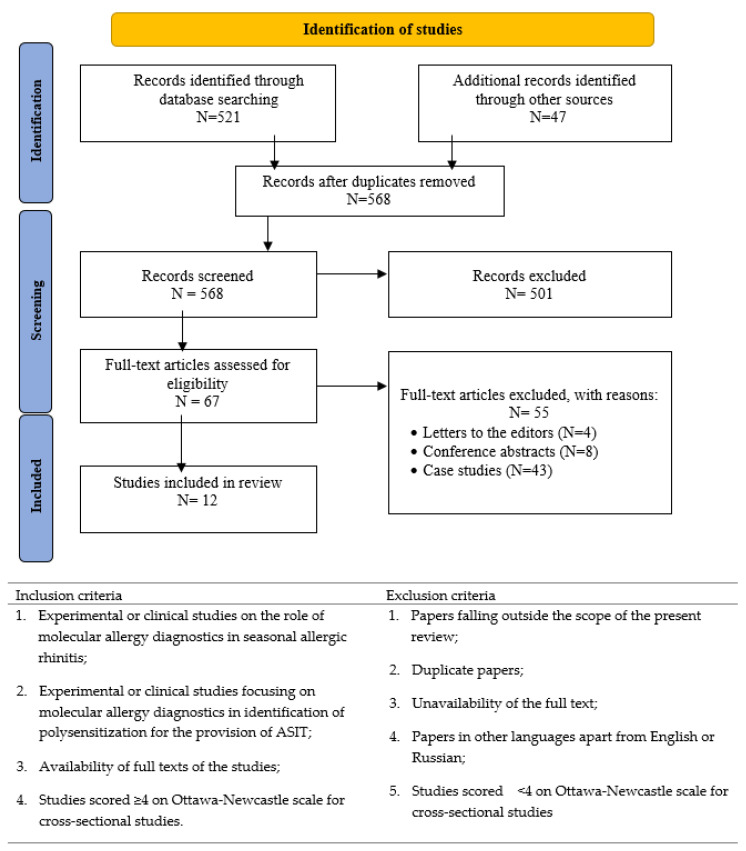
Study selection and eligibility criteria of systematic literature review.

**Figure 2 cells-12-00383-f002:**
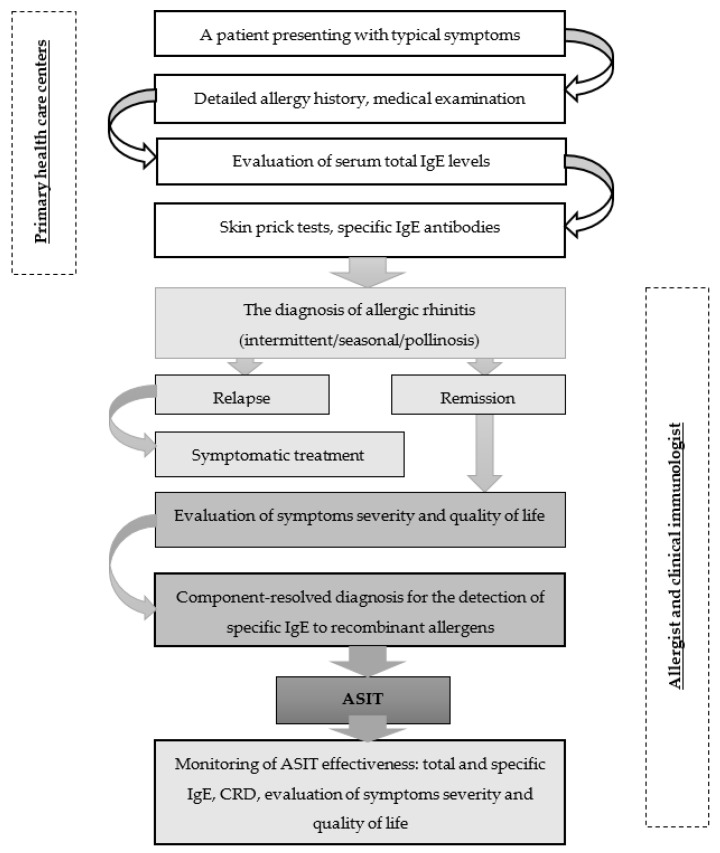
Management algorithm for patients with seasonal allergic rhinitis.

**Table 1 cells-12-00383-t001:** Studies evaluating the role of component-resolved diagnosis in providing allergen-specific immunotherapy to patients with seasonal allergic rhinitis.

Author, Year of Publication	Age of Patients	Sample Size	Type of Allergy	Country	Main Findings	Ottawa-Newcastle Score
Movérare, 2011 [72]	All ages	110	Mugwort allergy	Sweden, Estonia, Switzerland, Spain, Greece, United States, Canada	This research indicates a role for cross-reactive IgE antibodies in positive test results for mugwort in these subjects. Mugwort-sensitized subjects have different IgE reactivity profiles to weed allergens (Art v 3, Amb a 1, Par j 2, Sal k 1, Bet v 2, Bet v 4, CCDs). CRD in clinical practice leads to more detailed information on IgE reactivity at the molecular level that could be helpful to choose appropriate pollen extracts for ASIT *.	4
Sastre, 2012 [73]	All ages	141	Allergic rhinitis/asthma	Spain	There was a very low agreement between indications and use of allergens for specific immunotherapy before and after performing Component-resolved diagnosis (46% of patients).	4
Letrán, 2013 [74]	6–62 years	175	Seasonal pollen-allergic rhinoconjunctivitis and/or asthma	Spain	The use of Component-resolved diagnosis helped to carefully select allergens for ASIT. CRD ^#^ changed the selection of allergens for ASIT in more than 50% of cases, as compared with the baseline selection.	4
Moreno, 2014 [75]	5–65 years	1263	Seasonal allergic rhinitis, asthma, allergic conjunctivitis	Spain	The majority of patients (922 patients, 73%) would have been indicated for a mixture of grass and olive pollens for the provision of allergen immunotherapy. In 56.8% of patients, there was noncoincidence in the composition of allergen immunotherapy that would be selected before and after the investigation. CRD could help improve the selection of AIT in polysensitized patients.	4
Stringari, 2014 [76]	4–18 years	651	Allergic rhinitis, asthma	Italy	The decision on prescription or composition of specific immunotherapy was changed in 277 (42%) of 651 children or 315 (48%) of 651 children, depending on the European or American approach, respectively.	4
Darsow, 2014 [77]	>18 years	101	Allergic rhinoconjunctivitis	Germany	There was significant heterogeneity in molecular sensitization profiles. None of the patients exactly matched the allergen composition of previous specific immunotherapy, containing Phl p 1, Phl p 2, Phl p 5a/b, and Phl p 6, the selection of which was based on conjunctival and nasal provocation tests.	4
Schmid, 2016 [78]	All ages	24	Seasonal rhinoconjunctivitis	Denmark	CRD suggests a personalized approach to ASIT. Change in IgE and IgG_4_ levels may be used as an early biomarker for ASIT effectiveness.	7
Luo, 2017 [79]	All ages	346	Allergic rhinitis and/or asthma	China	Only 17.1% of patients were sensitized to major allergens Phl p 1 and Phl p 5, and 100.0% of patients were sensitized to cross-reactive component Phl p 4. The authors conclude that ASIT is not indicated to all patients with timothy grass pollen sensitization.	4
Martínez-Cañavate Burgos, 2018 [80]	5–18 years	281	Seasonal allergic rhinitis	Spain	Double sensitization to grass and olive pollen allergens was found in vitro in 76% of children for an IgE cutoff point of 0.35 kU/L. When the component-resolved diagnosis results were known, the composition of the prescribed immunotherapy was changed in 52.87% of cases.	4
Del-Río Camacho, 2018 [81]	8–12 years	70	Allergic rhinitis and/or asthma	Spain	CRD led to a modified immunotherapy prescription in 54.3% of patients. Indications to the single-allergen therapy increased from 18% to 51% when the component-resolved diagnosis was included. The decision to prescribe immunotherapy was reversed following component-resolved diagnosis in 9.3% of cases.	4
Hu, 2019 [82]	1–85 years	57	Allergic rhinitis, asthma	China	CRD identified the main dust mite allergen components leading to sensitization (nDer p 1, rDer p 2, nDer f 1, and rDer f 2) as well as cross-reactive components rDer p 10, which helped to make a meaningful selection of allergens for ASIT.	4
Haidar, 2021 [83]	All ages	83	Rhinitis, conjunctivitis, asthma	Romania	Most patients were polysensitized (62.65%), especially to other pollens, house dust mites, and animal danders. Only 90% of the patients with positive skin prick test to ragweed pollen extract also had increased specific serum IgE to Amb a 1.	4

ASIT *—allergen-specific immunotherapy, CRD ^#^—component-resolved diagnosis.

**Table 2 cells-12-00383-t002:** Main characteristics of inhalant allergens causing seasonal allergic rhinitis [84,85,86].

Allergen Type (Source)	Allergenic Molecule	Group Type	Marker Type	Cross-Reactive Allergens
Major Components of Airborne Allergens				
Grass pollens				
Bermuda grass	nCyn d 1	Group 1	Major allergen	Grass pollen
Timothy grass	rPhl p 1	Group 2	Major allergen	Grass pollen
	rPhl p 2	Group 2	Minor allergen	Grass pollen
	rPhl p 4	Berberine bridge enzyme	Minor allergen	Grass pollen
	rPhl p 5	Group 5	Major allergen	Grass pollen
	rPhl p 6	Group 6	Minor allergen	Grass pollen
	rPhl p 11	Ole-e-1-related protein	Minor allergen	Grass pollen
Tree pollens				
Birch	rBet v 1	PR-10 protein	Major allergen	Fruits, vegetables, nuts, seeds, beans, tree pollens (birch, alder, hazel, hornbeam), carrot, celery, apple, apricot, cherry, pear, spices
Japanese cedar	nCry j 1	Pectate lyase	Major allergen	Tree pollens
Cypress	nCup a 1	Pectate lyase	Major allergen	Tree pollens (cypress family: juniper, cypress, cedar)
Olive	rOle e 1	General olive group 5	Major allergen	Tree pollens It is a marker of a high degree of cross-reactivity with ash, privet, lilac, and angustifolia, although these pollens are not identical. rOle e1 is homologous with proteins of sycamore, plantain, saffron, and cereal crop: timothy grass, rye, and corn.
	rOle e 9	1,3-beta-glucanase	Minor allergen	Tree pollen
Platanus acerifolia, Plane tree	rPla a 1	Invertase inhibitor	Major allergen	Tree pollen
	rPla a 2	Polygalacturonase	Minor allergen	Tree pollen
Weed pollens				
Ambrosia	Amb a 1	Pectate lyase	Major allergen	Weed pollens It is a marker of true sensitization to ambrosia and cross-reactivity with cereal crop and weed pollens.
Artemisia vulgaris	Art v 1	Defensin	Major allergen	Weed, grass, and tree pollens. It is responsible for cross-reactivity with pollens of various plants: ragweed, daisy, chamomile, dandelion, sunflower, calendula, elecampane, string, coltsfoot, citrus fruits, kiwi, mango, sunflower seed, honey, chicory, parsley, carrots, tomatoes, peas, dill, hazelnuts, peanuts, red pepper.
Chenopodium album	rChe a 1	Ole-e-1-related protein	Major allergen	Weed pollens
Pellitory	rPar j 2	Lipid transport proteins (nsLTP)	Major allergen	Weed pollens
Plantain	rPla I 1	Ole-e-1-related protein	Major allergen	Weed pollens
Kali tragus	nSal k 1	Pectin methyl esterase	Major allergen	Weed pollens
Species-specific and cross-reactive components				
Olive pollen	Ole e 7	Lipid transport proteins (nsLTP)	Minor allergens	Fruits, vegetables, nuts, seeds, beans, cereal crop, spices, tree, and weed pollens
Plane tree	Pla a 3			
Birch	Bet v 1	PR-10 protein	Major allergens	Fruits, vegetables, nuts, seeds, beans, tree pollens (alder, hazel, hornbeam), carrot, celery, apple, peach, cherry, pear, spices, and peanuts.
Alder	Aln g 1			
Hazel	rCor a 1.0101			
Birch	rBet v 2	Profilin	Minor allergens	Fruits, vegetables, nuts, seeds, beans, cereal crop, spices, latex, weed, grass, and tree pollens (olive, bermudagrass, pellitory, sunflower, date fruit, banana, pineapple, and exotic fruits)
Forest grass	rMer a 1			
Timothy grass	rPhl p 12			
Birch	rBet v 4	Polcalcin	Minor allergens	Weed, grass, and tree pollens (Timothy grass, bermudagrass, turnip, rape, European olives, black alder) and could serve as a marker of polyvalent sensitization to plant allergens.
Timothy grass	rPhl p 7			Weed, grass, and tree pollens (beech family: birch and olives)

**Table 3 cells-12-00383-t003:** Classification of protein families in seasonal allergic rhinitis [55].

Group Type	Degree of Cross-Reactivity	Properties	Inhalant Allergens
Polcalcins (calcium-binding proteins)	High	-	Bet v4, Phl p7
Profilins	High	Susceptible to high temperatures and digestive enzymes	Bet v2, rMer a 1, Phl p12
Nonspecific lipid transport proteins (nsLTP)	Various	Resistant to high temperatures and digestive enzymes	Art v 3, Ole e 7, Pla a 3
Pathogenesis-related protein family 10 (PR-10), Bet v 1 homolog	High	Susceptible to high temperatures and digestive enzymes	Bet v 1, Aln g 1, rCor a 1.0101
Cross-reactive carbo-hydrate determinants (CCD)	High	Resistant to high temperatures	nCyn d1, nOle e1, nCup a 1, nSal k 1, nPla a 2, nArt v 1, Phl p 4

## Data Availability

The data presented in this study are available on request from the corresponding author.

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
