# Peer review of "Molecular Aspects of Allergen-Specific Immunotherapy in Patients with Seasonal Allergic Rhinitis"

_cells, 2023, doi:10.3390/cells12030383_

Round 1
Reviewer 1 Report
The paper addresses the role of molecular diagnostics in allergen-specific immunotherapy (AIT) by means of a systematic analysis. The main result of the study is that molecular diagnostics allows to reconsider the choice of AIT in some patients and the type of allergen to be administered in a significant proportion of patients.
Several major points should however be raised
The results of the systematic analysis, paragraph 3.4 Outcomes, should be expanded since this is the core point of the review.
The authors have based the analysis of molecular allergy diagnostics solely on a multiplex-based system, ISAC, Thermofischer. No mention is given to specific IgE with singleplex diagnostics for recombinants allergens, such as ImmunoCap, Thermofisher. The latter is a more widely adopted, less costly and quantitative method, as compared to the semiquantitative evaluation by ISAC. For the same reason, in Fig. 2, while molecular allergy diagnostic is correct, mentionining only ISAC test should be omitted.
The authors do not mention the role of cross-reactive carbohydrate determinants (CCD) which are usually responsible for extensive IgE cross-reactivity. The knowledge of sensitization to such allergens is very important and their presence often complicate the diagnostics and choice of allergen-specific immunotherapy and should be mentioned in the discussion.
Moreover, the authors do not mention other multiplex platforms such as ALEX, Allergy Explorer, which are theoretically less influenced by CCD.
In the discussion the author state that ASIT is “the only effective method available for the treatment of seasonal allergic rhinitis”. I would rephrase the sentence saying that is the only curative method available for the treatment of seasonal allergic rhinitis.[1]
Lipid transfer proteins are mainly food allergens, at least in Southern Europe, their role as inhalants is described only for Art v3 in China and Par a 3 in Spain, they should not be considered among the major cross-reactive inhalant allergens.
The title of Table 1 should be “ studies evaluating the role of “
In the study of Moverare and collegues the author mention Bet v1 but the allergen under study was Amb a 1. So the main finding comment is incorrect.
In table 2 some corrections should be made about allergen type and sources. More precisely, the authors cite Sycamore as the source of rPla a1. Perhaps it should be better to cite it as Platanus acerifolia, Plane tree. They state that Sagebrush is the source of Art v 1, perhaps it should be better to cite Artemisia vulgaris or ragweed. They also cite Quinoa as the allergen source of Art v3 (?).
In the discussion, paragraph 4.6, the authors cite ECP as a marker of effective of AIT. In reference 75, a EAACI study by Alvaro-Lozano no such statement exists. The role of ECP is stil not recognized and therefore should be also removed from Fig.2
The reference list fails to mention some important papers on AIT in allergic rhinitis treatment.
1. Matricardi PM, Dramburg S, Potapova E, Skevaki C, Renz H. Molecular diagnosis for allergen immunotherapy. J Allergy Clin Immunol. 2019 Mar;143(3):831-843.
2.Roberts G, Pfaar O, Akdis CA, Ansotegui IJ, Durham SR, Gerth van Wijk R, Halken S, Larenas-Linnemann D, Pawankar R, Pitsios C, Sheikh A, Worm M, Arasi S, Calderon MA, Cingi C, Dhami S, Fauquert JL, Hamelmann E, Hellings P, Jacobsen L, Knol EF, Lin SY, Maggina P, Mösges R, Oude Elberink JNG, Pajno GB, Pastorello EA, Penagos M, Rotiroti G, Schmidt-Weber CB, Timmermans F, Tsilochristou O, Varga EM, Wilkinson JN, Williams A, Zhang L, Agache I, Angier E, Fernandez-Rivas M, Jutel M, Lau S, van Ree R, Ryan D, Sturm GJ, Muraro A. EAACI Guidelines on Allergen Immunotherapy: Allergic rhinoconjunctivitis. Allergy. 2018 Apr;73(4):765-798. doi: 10.1111/all.13317.
3. Bousquet J, Pfaar O, Togias A, Schünemann HJ, Ansotegui I, Papadopoulos NG, Tsiligianni I, Agache I, Anto JM, Bachert C, Bedbrook A, Bergmann KC, Bosnic-Anticevich S, Bosse I, Brozek J, Calderon MA, Canonica GW, Caraballo L, Cardona V, Casale T, Cecchi L, Chu D, Costa E, Cruz AA, Czarlewski W, Durham SR, Du Toit G, Dykewicz M, Ebisawa M, Fauquert JL, Fernandez-Rivas M, Fokkens WJ, Fonseca J, Fontaine JF, Gerth van Wijk R, Haahtela T, Halken S, Hellings PW, Ierodiakonou D, Iinuma T, Ivancevich JC, Jacobsen L, Jutel M, Kaidashev I, Khaitov M, Kalayci O, Kleine Tebbe J, Klimek L, Kowalski ML, Kuna P, Kvedariene V, La Grutta S, Larenas-Linemann D, Lau S, Laune D, Le L, Lodrup Carlsen K, Lourenço O, Malling HJ, Marien G, Menditto E, Mercier G, Mullol J, Muraro A, O'Hehir R, Okamoto Y, Pajno GB, Park HS, Panzner P, Passalacqua G, Pham-Thi N, Roberts G, Pawankar R, Rolland C, Rosario N, Ryan D, Samolinski B, Sanchez-Borges M, Scadding G, Shamji MH, Sheikh A, Sturm GJ, Todo Bom A, Toppila-Salmi S, Valentin-Rostan M, Valiulis A, Valovirta E, Ventura MT, Wahn U, Walker S, Wallace D, Waserman S, Yorgancioglu A, Zuberbier T; ARIA Working Group. 2019 ARIA Care pathways for allergen immunotherapy. Allergy. 2019 Nov;74(11):2087-2102. doi: 10.1111/all.13805. Epub 2019 Jul 15. PMID: 30955224.
Author Response
|
Referee 1
|
Changes by the authors |
|||||
|
The paper addresses the role of molecular diagnostics in allergen-specific immunotherapy (AIT) by means of a systematic analysis. The main result of the study is that molecular diagnostics allows to reconsider the choice of AIT in some patients and the type of allergen to be administered in a significant proportion of patients. Several major points should however be raised |
Thank you for the thoughtful approach to evaluation of our manuscript and the suggestions made to improve its quality. We did our best to amend the manuscript to make it meet the publication requirements. All the changes made are highlighted in yellow. |
|||||
|
The results of the systematic analysis, paragraph 3.4 Outcomes, should be expanded since this is the core point of the review. |
3.4. Outcomes The outcomes of studies included in this review were assessed on the basis of ASIT effectiveness. The type of sensitization (monosensitization vs. polysensitization) and presence of cross-reactivity served as influence variables. According to the available evidence, these factors significantly influenced the change in allergen composition of ASIT. CRD use in clinical practice leads to more detailed information on IgE reactivity at the molecular level that could be helpful in the diagnosis of pulmonary allergy and might help clinicians choose appropriate pollen extracts for ASIT. (Table 1).
|
|||||
|
The authors have based the analysis of molecular allergy diagnostics solely on a multiplex-based system, ISAC, Thermofischer. No mention is given to specific IgE with singleplex diagnostics for recombinants allergens, such as ImmunoCap, Thermofisher. The latter is a more widely adopted, less costly and quantitative method, as compared to the semiquantitative evaluation by ISAC. For the same reason, in Fig. 2, while molecular allergy diagnostic is correct, mentionining only ISAC test should be omitted.
|
Done. We corrected the information presented in Figure 2. Also, we refer to Component-resolved diagnosis (CRD) throughout the manuscript. |
|||||
|
The authors do not mention the role of cross-reactive carbohydrate determinants (CCD) which are usually responsible for extensive IgE cross-reactivity. The knowledge of sensitization to such allergens is very important and their presence often complicate the diagnostics and choice of allergen-specific immunotherapy and should be mentioned in the discussion. |
Done. We included the following information into Tables 3: Table 3
Also, we added the following passage to the text of the Discussion section: Cross-reactive carbohydrate determinants (CCD) - source of positive IgE detection results without clinical significance, which could occur in the detection of specific IgE of plant allergens. In serum-based allergy diagnosis, antibodies of the IgE class directed against CCDs therefore give the impression of polysensitization. Anti-CCD IgE, however, does not seem to elicit clinical symptoms. Diagnostic results caused by CCDs are therefore regarded as false positives [55]. Some commercial tests, including CCD inhibitors such as ALEX, have been applied in clinical practice and have shown advantages in reducing false-positive IgE results without impacting diagnostic sensitivity toward relevant allergens.
|
|||||
|
Moreover, the authors do not mention other multiplex platforms such as ALEX, Allergy Explorer, which are theoretically less influenced by CCD. |
Done. Some commercial tests, including CCD inhibitors such as ALEX, have been applied in clinical practice and have shown advantages in reducing false-positive IgE results without impacting diagnostic sensitivity toward relevant allergens.
|
|||||
|
In the discussion the author state that ASIT is “the only effective method available for the treatment of seasonal allergic rhinitis”. I would rephrase the sentence saying that is the only curative method available for the treatment of seasonal allergic rhinitis.[1]
|
Done. We introduced the following correction: To date, ASIT is the only curative method available for the treatment of seasonal AR. |
|||||
|
Lipid transfer proteins are mainly food allergens, at least in Southern Europe, their role as inhalants is described only for Art v3 in China and Par a 3 in Spain, they should not be considered among the major cross-reactive inhalant allergens.
|
Done. We have deleted this information:
|
|||||
|
The title of Table 1 should be “ studies evaluating the role of “ |
Done. We changed the title of Table 1: Studies evaluating the role of component-resolved diagnosis in providing allergen-specific immunotherapy to patients with seasonal allergic rhinitis.
|
|||||
|
In the study of Moverare and collegues the author mention Bet v1 but the allergen under study was Amb a 1. So the main finding comment is incorrect. |
Done. We included the following information: This research indicates a role for cross-reactive IgE antibodies in positive test results for mugwort in these subjects. Mugwort-sensitized subjects have different IgE reactivity profiles to weed allergens (Art v 3, Amb a 1, Par j 2, Sal k 1, Bet v 2, Bet v 4, CCDs). CRD in clinical practice leads to more detailed information on IgE reactivity at the molecular level that could be helpful to choose appropriate pollen extracts for ASIT*.
|
|||||
|
In table 2 some corrections should be made about allergen type and sources. More precisely, the authors cite Sycamore as the source of rPla a1. Perhaps it should be better to cite it as Platanus acerifolia, Plane tree. They state that Sagebrush is the source of Art v 1, perhaps it should be better to cite Artemisia vulgaris or ragweed. They also cite Quinoa as the allergen source of Art v3 (?).
|
You are right. We cited Sycamore as Platanus acerifolia, Plane tree; indicated that the source of Art v 1 is Artemisia vulgaris; and replaced Quinoa with Chenopodium album. |
|||||
|
In the discussion, paragraph 4.6, the authors cite ECP as a marker of effective of AIT. In reference 75, a EAACI study by Alvaro-Lozano no such statement exists. The role of ECP is stil not recognized and therefore should be also removed from Fig.2
|
Done. We removed ECP from Discussion and from Fig.2.
|
|||||
|
The reference list fails to mention some important papers on AIT in allergic rhinitis treatment. 1. Matricardi PM, Dramburg S, Potapova E, Skevaki C, Renz H. Molecular diagnosis for allergen immunotherapy. J Allergy Clin Immunol. 2019 Mar;143(3):831-843.
2.Roberts G, Pfaar O, Akdis CA, Ansotegui IJ, Durham SR, Gerth van Wijk R, Halken S, Larenas-Linnemann D, Pawankar R, Pitsios C, Sheikh A, Worm M, Arasi S, Calderon MA, Cingi C, Dhami S, Fauquert JL, Hamelmann E, Hellings P, Jacobsen L, Knol EF, Lin SY, Maggina P, Mösges R, Oude Elberink JNG, Pajno GB, Pastorello EA, Penagos M, Rotiroti G, Schmidt-Weber CB, Timmermans F, Tsilochristou O, Varga EM, Wilkinson JN, Williams A, Zhang L, Agache I, Angier E, Fernandez-Rivas M, Jutel M, Lau S, van Ree R, Ryan D, Sturm GJ, Muraro A. EAACI Guidelines on Allergen Immunotherapy: Allergic rhinoconjunctivitis. Allergy. 2018 Apr;73(4):765-798. doi: 10.1111/all.13317.
3. Bousquet J, Pfaar O, Togias A, Schünemann HJ, Ansotegui I, Papadopoulos NG, Tsiligianni I, Agache I, Anto JM, Bachert C, Bedbrook A, Bergmann KC, Bosnic-Anticevich S, Bosse I, Brozek J, Calderon MA, Canonica GW, Caraballo L, Cardona V, Casale T, Cecchi L, Chu D, Costa E, Cruz AA, Czarlewski W, Durham SR, Du Toit G, Dykewicz M, Ebisawa M, Fauquert JL, Fernandez-Rivas M, Fokkens WJ, Fonseca J, Fontaine JF, Gerth van Wijk R, Haahtela T, Halken S, Hellings PW, Ierodiakonou D, Iinuma T, Ivancevich JC, Jacobsen L, Jutel M, Kaidashev I, Khaitov M, Kalayci O, Kleine Tebbe J, Klimek L, Kowalski ML, Kuna P, Kvedariene V, La Grutta S, Larenas-Linemann D, Lau S, Laune D, Le L, Lodrup Carlsen K, Lourenço O, Malling HJ, Marien G, Menditto E, Mercier G, Mullol J, Muraro A, O'Hehir R, Okamoto Y, Pajno GB, Park HS, Panzner P, Passalacqua G, Pham-Thi N, Roberts G, Pawankar R, Rolland C, Rosario N, Ryan D, Samolinski B, Sanchez-Borges M, Scadding G, Shamji MH, Sheikh A, Sturm GJ, Todo Bom A, Toppila-Salmi S, Valentin-Rostan M, Valiulis A, Valovirta E, Ventura MT, Wahn U, Walker S, Wallace D, Waserman S, Yorgancioglu A, Zuberbier T; ARIA Working Group. 2019 ARIA Care pathways for allergen immunotherapy. Allergy. 2019 Nov;74(11):2087-2102. doi: 10.1111/all.13805. Epub 2019 Jul 15. PMID: 30955224. |
Done. We included the following references:
53. Matricardi, P.M.; Dramburg, S.; Potapova, E.; Skevaki, C.; Renz, H. Molecular Diagnosis for Allergen Immunotherapy. J. Allergy Clin. Immunol. 2019, 143, 831–843, doi:10.1016/j.jaci.2018.12.1021.
92. Roberts, G.; Pfaar, O.; Akdis, C.A.; Ansotegui, I.J.; Durham, S.R.; Gerth van Wijk, R.; Halken, S.; Larenas-Linnemann, D.; Pawankar, R.; Pitsios, C.; et al. EAACI Guidelines on Allergen Immunotherapy: Allergic Rhinoconjunctivitis. Allergy Eur. J. Allergy Clin. Immunol. 2018, 73, 765–798, doi:10.1111/all.13317.
48. Bousquet, J.; Pfaar, O.; Togias, A.; Schünemann, H.J.; Ansotegui, I.; Papadopoulos, N.G.; Tsiligianni, I.; Agache, I.; Anto, J.M.; Bachert, C.; et al. 2019 ARIA Care Pathways for Allergen Immunotherapy. Allergy Eur. J. Allergy Clin. Immunol. 2019, 74, 2087–2102, doi:10.1111/all.13805. |
Reviewer 2 Report
This article involves with systematic review and 'narrative synthesis' of molecular allergy diagnosis (MAD) for allergen specific immunotherapy (AIST) in seasonal allergic rhinitis (SAR).
The methodology is good especially clear search terms, clear inclusions and exclusion criteria, clear scoring method according to the PRISMA guideline.
However, there are some points that should be addressed for improving the pattern of writing style of systematic review article.
1. In the introduction part (page #2), the authors should discuss more about the issue about the role of MAD in single-allergen ASIT vs multi-allergen ASIT in 'polysensitized SAR. Actually, some content in the discussion part (line#51 to line#164) can be moved into the introduction part.
2. The key finding from this study is 'MAD changes the choice of relevant allergens for ASIT in at least 50% of SAR'. The authors already described in the discussion part and elaborate their idea in Line#191 to #260. So, the finding from this systematic review (in the result part) should be linked with the discussion part.
Author Response
|
Referee 2 |
Changes by the authors |
|
This article involves with systematic review and 'narrative synthesis' of molecular allergy diagnosis (MAD) for allergen specific immunotherapy (AIST) in seasonal allergic rhinitis (SAR). The methodology is good especially clear search terms, clear inclusions and exclusion criteria, clear scoring method according to the PRISMA guideline. However, there are some points that should be addressed for improving the pattern of writing style of systematic review article.
|
Thank you for the thoughtful approach to evaluation of our manuscript and the suggestions made to improve its quality. We did our best to amend the manuscript to make it meet the publication requirements. All the changes made are highlighted in yellow.
|
|
1. In the introduction part (page #2), the authors should discuss more about the issue about the role of MAD in single-allergen ASIT vs multi-allergen ASIT in 'polysensitized SAR. Actually, some content in the discussion part (line#51 to line#164) can be moved into the introduction part.
|
You are right. We moved a content (line#51 to line#136) from the Discussion part to the Introduction part: To date, ASIT is the only curative method available for the treatment of seasonal AR. This method impacts all pathogenic mechanisms of allergy, potentiates preventive effects and enables lasting remission after the end of treatment. According to Allergic Rhinitis and its Impact on Asthma (ARIA) guidelines that were initiated at the WHO workshop, ASIT is a disease-modifying therapy with grade IА recommendation. The clinical effects of ASIT include reduction of AR symptoms, decreased demand for anti-allergic agents, prevention of the expansion of allergen spectrum and bronchial asthma in patients with seasonal AR. Finally, ASIT improves patient quality of life. The method is based on the systematic introduction of causative allergens in gradually increasing dosages. This leads to specific hyposensitization, which promotes immune tolerance to the causative allergen. In keeping with international guidelines, ASIT has to be provided for 3–5 years to achieve stable results, although the meaningful effect is already visible after the first year of treatment. At present, ASIT is available in two forms: subcutaneous immunotherapy (SCIT) and sublingual immunotherapy (SLIT). Allergen-specific subcutaneous immunotherapy was developed in 1908 and has since gained much recognition worldwide. However, the application of allergen-specific sublingual immunotherapy caused many doubts due to unpurified vaccines, and it became widespread only within the past two decades. SLIT has several advantages compared with SCIT. First, it could be used in children from the age of five. Second, only the first intake of medication has to be made in the presence of a doctor, and other intakes could be made independently at home with only periodic visits to a doctor for follow-up. Third, when medication comes in tablet form, it causes less discomfort for a patient, in contrast with repeated injections, which may be quite stressful. In addition, SLIT has a better safety profile with only local side effects and an extremely low incidence of anaphylaxis and other life-threatening conditions, often associated with treatment discontinuation. Thus, ASIT is currently considered the preferred treatment for allergic disorders and has sufficient proof of clinical effectiveness.
|
|
2. The key finding from this study is 'MAD changes the choice of relevant allergens for ASIT in at least 50% of SAR'. The authors already described in the discussion part and elaborate their idea in Line#191 to #260. So, the finding from this systematic review (in the result part) should be linked with the discussion part. |
You are right. Done:
The problem with polysensitized patients is that they are sensitized to multiple sources of allergens, including cross-reactive ones. In Moreno et al.'s study, 78.2% of the subjects had polysensitization, while the rest were monosensitized or had a negative result. Hu et al. showed that more than 93.0% of subjects were sensitive to more than one allergen component. Haidar et al. reported that the majority of patients were polysensitized (62.65%). Therefore, the question of prescribing multiple allergens for these patients with ASIT and how effective this treatment will be has long been discussed. Calderon et al. concluded that single-dose therapy with two unrelated allergens is clinically effective. Darsow et al. reported that CRD would be helpful in deciding on the indication for conventional ASIT in polysensitized patients. In addition, Movérare et al. revealed that CRD use in clinical practice leads to more detailed information on IgE reactivity at the molecular level that could be helpful in the diagnosis of polyclonal allergy and might help clinicians choose appropriate pollen extracts for ASIT. Many authors report the importance of using CRD to select and influence the effectiveness of ASIT. Moreno et al. argue that when selecting the composition of AIT based solely on the results of SPT (skin prick test), approximately one-third of the patients would be treated with an allergen to which they were not allergic. Thus, according to the results of CRD, more than half of the patients had a change in the composition of ASIT compared to the initial selection. Sastre et al. observed that the composition of AIT before and after molecular diagnosis coincided only in 46% of patients. Letrán et al. reported a change in the ASIT protocol in more than 50% of cases after using CRD. Moreover, Del-Río Camacho et al. described in their study that protocols for ASIT were changed in 54.3% of cases after CRD was conducted. Martinez-Cañavate et al. reported that specialists changed the composition of the prescribed immunotherapy in 52.87% of cases. In addition, Schmid et al. found that CRD might be a useful companion diagnostic to monitor the efficacy of SCIT. These facts confirm the need to use CRD when selecting the composition of allergens for AIT. |
Round 2
Reviewer 1 Report
In the introduction, the authors cite primary allergens. Perhaps it would be better to use the terms major allergens or allergens associated with a genuine sensitization to a given allergenic source or specific allergen component.
The authors describe the advantages of SLIT over SCIT. Why not describing the advantages of SCIT over SLIT?
“Certain studies failed to establish a relationship between specific IgE and the effectiveness of ASIT, while others demonstrated some deterioration of specific IgE with ASIT”. This sentence is not clear and lacks a reference.
Not all allergens used in CRD are recombinant proteins.
Abbreviations in parentheses, such as CRD, ASIT, AR, should be used only after mentioning the full term in the text and not vice versa, please check the manuscript. CRD is not mentioned in extenso in the text before its abbreviation is used.
Despite some improvement, there are several concerns about how the results and discussion are structured.
3.1. Patients’ characteristics. In this section, in addition to patient age, the authors should also describe how patients were selected e.g., through skin prick test/specific IgE and clinical history, or purely IgE sensitization or skin prick test. Were patients mono- or polysensitized?
3.3. Aims of CRD in ASIT. The authors should clearly state which are the outcomes of interest after CRD, primary and secondary, for example, ever prescribing ASIT, changing allergen source, switching from a mixed preparation to a single source, concordance of diagnostics, before and after CRD and so on.
3.4. Outcomes. I believe the outcome here should be diagnostic accuracy for immunotherapy prescription. However, an interesting point to comment in the discussion is whether the choice of ASIT based of CRD is corroborated by studies evaluating clinical symptoms scales, allergen exposures in chambers or specific IgE and IgG or other biomarkers after ASIT.
As to the conclusion, its structure is not consequential and difficult to understand. The management algorithm is interesting. Its description should be expanded in the text.
The authors mention here a list of results which should better suit the results’ paragraph. Citing cross-reactive food allergens is confusing in this instance.
The expression “allergens need to be selected by CRD technology” is unclear.
The discussion should focus more of the advantages of CRD for ASID and implications for diagnostic management and touching the strengths and limitations of available studies, as suggested in 3.4.
Author Response
Response to Reviewer 1 Comments
Thank you for the thoughtful approach to evaluation of our manuscript and the suggestions made to improve its quality. We did our best to amend the manuscript to make it meet the publication requirements. All the changes made are highlighted in yellow.
Point 1: In the introduction, the authors cite primary allergens. Perhaps it would be better to use the terms major allergens or allergens associated with a genuine sensitization to a given allergenic source or specific allergen component.
Response 1: We have replaced the word primary with major allergens.
Point 2: The authors describe the advantages of SLIT over SCIT. Why not describing the advantages of SCIT over SLIT?
Response 2: Nonetheless, when we focus on seasonal diseases, comparative data show that SCIT has a higher efficacy [39].
Point 3: “Certain studies failed to establish a relationship between specific IgE and the effectiveness of ASIT, while others demonstrated some deterioration of specific IgE with ASIT”. This sentence is not clear and lacks a reference.
Response 3: A relationship between specific IgE and the effectiveness of ASIT may be attributed to specific IgG, which is the marker of ASIT effectiveness.
Point 4: Not all allergens used in CRD are recombinant proteins.
Response 4: Done. We included the following information in Table 2 and Table 3: Group type
Point 5: Abbreviations in parentheses, such as CRD, ASIT, AR, should be used only after mentioning the full term in the text and not vice versa, please check the manuscript. CRD is not mentioned in extenso in the text before its abbreviation is used.
Response 5: Done. We corrected the information
Point 6: Despite some improvement, there are several concerns about how the results and discussion are structured.
3.1. Patients’ characteristics. In this section, in addition to patient age, the authors should also describe how patients were selected e.g., through skin prick test/specific IgE and clinical history, or purely IgE sensitization or skin prick test. Were patients mono- or polysensitized?
Response 6: Patients were selected through skin prick test and clinical history in 7 studies [72–78], specific IgE and clinical history in 1 study [79], provocation tests and clinical history in 2 studies [80,81] and purely IgE sensitiza-tion or skin prick test in 2 studies [82,83]. Patients were monosensitized in 3 studies [76,80,82], polysensitized in 2 studies [77,81], mono- and polysensitized in 7 studies [72–75,78,79,83].
Point 7: 3.3. Aims of CRD in ASIT. The authors should clearly state which are the outcomes of interest after CRD, primary and secondary, for example, ever prescribing ASIT, changing allergen source, switching from a mixed preparation to a single source, concordance of diagnostics, before and after CRD and so on.
Response 7: Type of Participants: Studies of patients without a prior confirmation of the type of sensitization were not eligible for inclusion.
Type of Exposure: CRD confirms the choice of allergen indication and use for ASIT.
Type of Outcome: The primary outcome was the decision on the cor-rectness of the ASIT protocol and diagnostic accuracy for immunotherapy prescription. Secondary outcomes included identifying allergen source types, changing allergen source, and diagnostic concordance before and after CRD.
Study Design: Observational studies, such as case reports and case se-ries, case-control studies, randomized control trials and prospective and retrospective cohort studies were eligible for inclusion.
Point 8: 3.4. Outcomes. I believe the outcome here should be diagnostic accuracy for immunotherapy prescription. However, an interesting point to comment in the discussion is whether the choice of ASIT based of CRD is corroborated by studies evaluating clinical symptoms scales, allergen exposures in chambers or specific IgE and IgG or other biomarkers after ASIT.
Response 8: The outcomes of studies included in this review were assessed on the basis of diagnostic accuracy of CRD for immunotherapy prescription. Influence variables included allergenic molecule, marker type, and type of sensitization (monosensitization vs. polysensitization).
Point 9: As to the conclusion, its structure is not consequential and difficult to understand. The management algorithm is interesting. Its description should be expanded in the text. The authors mention here a list of results which should better suit the results’ paragraph. Citing cross-reactive food allergens is confusing in this instance.
Response 9: Identification of potential ASIT patients with typical symptoms should begin at primary health care centers and include a detailed allergy history, medical examina-tion, and assessment of serum total IgE levels to confirm allergic sensitization.
Medical professionals can clearly understand the diagnostic trajectory of individ-uals with suspected allergic rhinitis by using this algorithm. Furthermore, it allows for the reduction of overdiagnosis errors during the primary medical care stage. This algo-rithm also serves as a guide for allergology experts, as it incorporates current advice on cutting-edge technology for allergy diagnosis and treatment.
Conclusions
Modern allergology services prioritize the reduction of seasonal allergic rhi-nitis rates. CRD is highly relevant in this regard for the diagnostic accuracy of CRD for immunotherapy prescription. Such an integrated approach is most ap-propriate for assessing major and cross-reactive allergen components as well as identifying cases of polysensitization. Besides Furthermore, the search for prognostic biomarkers that can predict the efficacy of ASIT must be continued.
Point 10: The expression “allergens need to be selected by CRD technology” is unclear.
Response 10: Done. We corrected this information: set of allergens for ASIT need to be selected by CRD technology
Point 11: The discussion should focus more of the advantages of CRD for ASID and implications for diagnostic management and touching the strengths and limitations of available studies, as suggested in 3.4.
Response 11: The review's strengths are that we investigated the impact of several variables on treatment outcomes, including allergenic molecule type, marker type, and sensitization type, and discovered that they did have a significant impact on the ASIT protocol. This study describes the allergen's major and cross-reactive components at the molecular level, which can aid in the diagnosis of respiratory allergy and help clinicians select appropriate pollen extracts for ASIT.
This review has several limitations. The most significant limitations are financial, as CRD is an expensive method of research that may not be available to all patients. A limitation of some studies is the small number of participants, different sensitization profiles, and a different disease profile when allergic rhinitis is combined with bronchial asthma. In addition, different age profiles were included in some of the studies.
